# Effect of Cationic Lipid Nanoparticle Loaded siRNA with Stearylamine against Chikungunya Virus

**DOI:** 10.3390/molecules27041170

**Published:** 2022-02-09

**Authors:** Manish Kumar Jeengar, Mallesh Kurakula, Poonam Patil, Ashwini More, Ramakrishna Sistla, Deepti Parashar

**Affiliations:** 1ICMR-National Institute of Virology, 20-A, Dr. Ambedkar Road, Pune 411001, Maharashtra, India; manishkumarj@pharmacy.aims.amrita.edu (M.K.J.); poonamshewale07@rediffmail.com (P.P.); ashwini05.s@gmail.com (A.M.); 2Department of Pharmacology, Amrita School of Pharmacy, Amrita Vishwa Vidyapeetham, AIMS Health Sciences Campus, Kochi 682041, Kerala, India; 3CSIR-Indian Institute of Chemical Technology [CSIR-IICT], Hyderabad 500007, Telangana, India; mallesh_kurakula@yahoo.com; 4Product Development, CURE Pharmaceutical, Oxnard, CA 93033, USA

**Keywords:** chikungunya, non-viral vectors, cationic lipids, siRNA, nanodelivery systems, stearylamine

## Abstract

Chikungunya is an infectious disease caused by mosquito-transmitted chikungunya virus (CHIKV). It was reported that NS1 and E2 siRNAs administration demonstrated CHIKV inhibition in in vitro as well as in vivo systems. Cationic lipids are promising for designing safe non-viral vectors and are beneficial in treating chikungunya. In this study, nanodelivery systems (hybrid polymeric/solid lipid nanoparticles) using cationic lipids (stearylamine, C9 lipid, and dioctadecylamine) and polymers (branched PEI-g-PEG -PEG) were prepared, characterized, and complexed with siRNA. The four developed delivery systems (F1, F2, F3, and F4) were assessed for stability and potential toxicities against CHIKV. In comparison to the other nanodelivery systems, F4 containing stearylamine (Octadecylamine; ODA), with an induced optimum cationic charge of 45.7 mV in the range of 152.1 nm, allowed maximum siRNA complexation, better stability, and higher transfection, with strong inhibition against the E2 and NS1 genes of CHIKV. The study concludes that cationic lipid-like ODA with ease of synthesis and characterization showed maximum complexation by structural condensation of siRNA owing to high transfection alone. Synergistic inhibition of CHIKV along with siRNA was demonstrated in both in vitro and in vivo models. Therefore, ODA-based cationic lipid nanoparticles can be explored as safe, potent, and efficient nonviral vectors overcoming siRNA in vivo complexities against chikungunya.

## 1. Introduction

Chikungunya (CHIK) is a mosquito-borne disease caused by chikungunya virus (CHIKV). The resurgence of CHIK in the Indian Ocean Islands and India drew worldwide attention in 2005–2006 due to its explosive nature [1]. CHIKV is a positive-sense single-stranded RNA virus that is 11.8 kb long, belonging to the alphavirus genus and the Togaviridae family. Chikungunya fever is usually associated with headache, joint pain, rash, and arthralgia. Humans have become a reservoir of CHIKV for mosquitoes [2]. As per the National Vector-Borne Disease Control Program (NVBDCP), the epidemiology profile of chikungunya fever in India since its reemergence in 2005–2006 is alarming [3]. The majority of the initiatives to combat this disease are preventive rather than therapeutic. This is largely due to the lack of specific prophylactic or effective therapeutics available in the market. There is a strong urge to addresses this unmet need [4].

Gene therapy has gained enormous attention as a novel therapeutic approach for diversified pathologies including neoplastic, genetic, and infectious diseases. Therapy includes a functional procedure to treat or alleviate disease by genetically amending the cell of the patient [5]. RNA interference (RNAi) regulates both gene expression and cellular-defense mechanisms against viruses, especially in mammalian cells. Small interfering RNAs (siRNA) are pivotal in RNAi, a process of sequence-specific gene silencing without activating an interferon response in animal cells. This discovery has opened up newer technologies for molecular targeting of various cancer and viral infections [6]. Earlier, we have designed, screened, and evaluated the therapeutic potential of two promising siRNAs (NS1 and E2) against CHIKV in a mouse model [7]. However, the siRNA degradation by endogenous nucleases minimizes their half-life, and the smaller size aids rapid clearance by the kidney, limiting their in vivo efficacy [8]. Thus, there is a need for a stable, inert system that can efficiently encapsulate and transfect into the cell, allowing an effective siRNA-induced knockdown with minimal degradation or rapid in vivo clearance. Designing a stealth deliverables system to bypass the reticuloendothelial system for siRNA is quite promising for better targeting, sustained-release minimizing the dosage regimen, and unwanted cytotoxicity [9].

An efficient delivery system is essential for effective gene therapy. Both viral- and non-viral-based vectors have been evaluated to date [10]. Viral vectors are commonly used because of high transfection efficiency, but due to their low carrying capacity, immunogenicity, limited specificity, and unwanted cytotoxicity, their use has been limited [11]. The development of nucleic-acid-based therapy for chikungunya has been hampered due to methodological delivery limitations and undesirable side effects. These restrictions can be addressed with advances in the nanotechnology fields with the development of a nanoparticle-based siRNA complex as a non-viral gene delivery system. High biocompatibility, low cytotoxicity, and cost favor their use in comparison to viral vectors [12,13].

One of the best designated and optimized delivery systems among non-viral vectors for gene therapy are cationic solid lipid nanoparticles (SLN) [14]. SLN are easily fabricated using biocompatible materials, owing to their safety with better nucleic-acid encapsulation, stability, and cellular uptake [15]. The ease of surface trimming of SLN for site-specific targeting and extended blood circulation time makes them an ideal choice for gene delivery [16]. Presently, there is one FDA-approved, siRNA-based therapy for treatment of hATTR amyloidosis [17].

Cationic lipids (amphiphilic molecules) have gained significance in recent years due to their ease of synthesis, are extensively characterized, and can enable the elucidation of structure–activity relationships by functional domains’ modifications [18]. The polar head group of the cationic lipid enables the condensation of nucleic acid by electrostatic interactions with the phosphate groups of genes having a negative charge and further regulates the transfection efficiency [19]. Cationic lipids are categorized based on the nature and charged density of hydrophilic head groups. Slight modifications in the head groups can help to bypass varied in vitro and in vivo transfection barriers [20]. Stearylamine/octadecylamine (ODA) is a cationic lipid that has not yet been fully explored for its intrinsic properties and for its role in formulating drug or gene deliverables [21]. It was reported for the uniform distribution of cationic charge in and around the nanocarrier formulations like liposome even at the transmembrane pH gradient [22]. The nitrogen hydrophilic head group of ODA has a strong potential to become cytotoxic by interacting with critical enzymes like PKC along with good transfection [23]. It has even been reported for its anti-helminthic property [24], and its activities can be altered by other neutral or helper lipids that can be beneficial [25]. Therefore, we utilized ODA to induce a cationic charge in lipid nanoparticles for better complexation and transfection of siRNA.

In the present study, cationic charged systems for the effective complexation and delivery of siRNA against CHIKV using various lipids were developed, which was not reported earlier. The four formulated delivery systems were characterized and evaluated in the in vitro system for their potential in catering two different siRNAs, i.e., CHIK1 and CHIK 5 (designed against the ns1 and the E2 gene region) alone or in mixed form to inhibit the CHIKV replication. The optimized siRNA-based delivery system was further assessed for its safety, stability, and efficacy. Among the four different delivery systems developed, we inform about utilizing and evaluating the synergistic or antagonistic or additive effect of ODA in formulating delivery system against CHIKV for the good transfection of siRNA, with retained functionality.

## 2. Results

### 2.1. Formulation Development and Optimization of Delivery Systems

Different formulations were prepared using various cationic lipids to evaluate and rationalize an optimum delivery system. F1, F2, and F3 delivery systems prepared using cationic polymers indicated low complexation with siRNA in comparison to F4. Octadecylamine used in the preparation of F4 delivery systems indicated the maximum entrapment of siRNA. Additionally, in our previous studies, pure ODA (8 mg) alone has indicated strong inhibition of CHIKV in vitro and in vivo. Therefore, we anticipated that the delivery system prepared with ODA would exhibit a strong inhibition of CHIKV, and it was further optimized.

### 2.2. Particle Size, Polydispersity, and Zeta-Potential Determination

The average nanoparticle size, the polydispersity index (PDI), and the surface charges of the F1, F2, F3, and F4 formulations were determined and recorded (Figure 1). Among the cationic lipids used, the F4 formulation indicated a maximum cationic charge density (45.7 ± 4.2) in comparison to F1, F2, and F3, which had weak cationic charges. F4 indicated an average size (152.4 ± 0.36) and polydispersity of nanoparticles (<0.5) in comparison to other delivery systems. All formulations were filtered through a 0.45 µM filter before performing further studies, i.e., an MTT assay and conjugation with siRNA.

### 2.3. Evaluation of Optimal Lipid to siRNA Complexation Ratio for F4

As a part of the F4-formulation optimization and in order to assess the critical lipid concentration for maximum complexation of siRNA, the role of change in both particle size and zeta potential depending on the N/P ratio was determined (Figure 2). The experiments revealed that the size of the lipid nanoparticles increased with siRNA complexation, and the surface charge varied depending upon the lipid concentration used. However, our studies revealed that increasing the cationic lipid concentration did not exhibit a logarithmic increase in cationic charge density over nanoparticles. For example, for an increase from 2 to 8 mg of ODA, we observed a maximum of 45.7 ± 4.2 mV for the surface charge, and a further increase in the ODA concentration to 10 mg did not increase the cationic charge proportionally. With the direct-method complexation of cationic lipid with siRNA, the particle size was seen at a maximum at an N/P ratio of 6, and the surface charge increased from 20 to 45 mV as the N/P ratio increased from 1 to 8.

### 2.4. In Vitro Release Studies

The siRNA release was performed in two different pH conditions (pH 7.4 and 6.4). The results indicated that at the neutral condition the percent release of siRNA was higher in comparison to the acidic condition. There was a burst effect observed in the acidic condition that was minimized indicating a prolonged release in neutral conditions (Figure 3).

### 2.5. In Vitro Cytotoxicity Studies

The effect of different delivery systems F1, F2, F3, and F4 on the Vero cells in the culture was studied using an MTT assay. Based on these findings, the developed delivery systems F3 and F4, having a CC-50 value >100 µg/mL, were considered non-toxic to Vero cells, and both were further selected for preliminary screening for an optimal and a suitable delivery vehicle for siRNA in treating CHIK infection (Figure 4).

### 2.6. Optimization of siRNA Effective Concentration

To determine the optimal antiviral concentration of siRNA, the Vero cells were infected with the CHIKV and after that transfected with different concentrations of siRNA Chik-1 ranging from 200 pmol to 2000 pmol using lipofectamine reagent to the culture medium 1 h after infection and incubation were done for 24 h. The culture supernatant was evaluated for viral genomic RNA levels using real-time RT-PCR. The results show that rgw concentration equal to or greater than 400 pmol concentration showed a significant reduction in viral load (Figure 5A), so we selected the lowest conc. i.e., 400 pmol.

### 2.7. Transfection Efficiency Comparison between F3 and F4

siRNA (Chik1 concentration 400 pmol) loaded with F3 and F4 delivery systems was transfected to Vero cells after CHIKV infection. No significant decrease in viral load was observed in Chik-1 loaded with the F3-delivery-system-treated cells compared to viral control cells (Figure 5B). The Chik-1-loaded F4 delivery system showed a significant decrease in viral load compared to VC- (*p* < 0.001) and F3 + Chik1- (*p* < 0.01) treated cells. F4 + chik1-treated cells showed a significant decrease in viral load; hence, the F4 delivery system was selected over F3, for further studies (Figure 5B).

### 2.8. Determination of siRNA Complex with F4 SLN Using Agarose Gel Electrophoresis

As depicted in Appendix A, the siRNA movement was retarded in the ODA-loaded F4 siRNA delivery system in comparison to the naked siRNA at 4 °C and 37 °C temperatures. Therefore, the studies strongly indicate that ODA formed a stable complex and resulted in strong complexation with siRNA.

### 2.9. In Vitro Anti-CHIKV Activity of the siRNA-loaded F4 SLN Delivery System

siRNA Chik 1 and Chik-5, at individual and combination levels (400 pmol), were transfected using lipofectamine reagent and incubated for 24 h and 48 h post-treatment. In a similar way, siRNAs were loaded with the F4 SLN delivery system and incubated with CHIKV-infected Vero cells for 24 h and 48 h. The RT-PCR results showed that the siRNA complexed with the F4 delivery system significantly decreased the viral load at both time points; in the case of the Chik 1 and Chik 5 combination delivered with F4, the viral-load inhibition was greater when compared with that transfected with lipofectamine (Figure 6A,B). A decreased viral load at 24 h by the treatment of Chik 1 + F4 SLN was significantly (*p* < 0.01) greater compared to Chik 1-treated cells; similarly, Chik1 + Chik 5 + F4 was significantly (*p* < 0.001) more effective when compared to Chik 1 + Chik 5-treated cells (Figure 6A). Figure 6B showsthat all siRNA + F4-treated cells showed more significant (*p* < 0.001) inhibition of viral load compared to their respective naked siRNA-treated cells. The FFU assay also showed a 100% reduction using siRNA Chik1 and Chik5 loaded with the F4 SLN delivery system at 24 h and 48 h (Figure 6C,D) These results were also confirmed with immunofluorescence assay, and even the percentage of infected cells also showed the same effect (Figure 6E,F).

### 2.10. In Vivo Anti-CHIKV Activity of siRNA-loaded F4 SLN Delivery System

To investigate the anti-CHIKV activities of the siRNA-loaded F4 delivery system, in vivo female C57BL/6 mice (4–5 weeks) were used. The mice were intramuscularly (I.M.) inoculated with 100 µL CHIKV, i.e., 10^7^ pfu on day 0. siRNA Chik 1, Chik 5, and Chik1 + Chik 5 of both were incubated with an F4 delivery system and administered intravenously on the 3rd day. Blood collection was done on days 3, 5, and 7 for viremia analysis. Muscle tissue was collected from the hind feet of mice, and histopathology and RT-PCR analysis was done at different time points. Treatment with siRNA complexed with F4 SLN exhibited a significant reduction in the serum viral load as evaluated at days 5 and 7 using real-time RT-PCR. The Chik-1 and Chik-5 siRNA combination delivered with F4 showed a ~99% reduction in viremia compared to the control group at days 5 and 7 in both serum (Figure 7A) and muscle tissue (Figure 7B). Histopathology results (Figure 8) also indicated that treatment of the siRNA F4 complex reduced the inflammatory cell infiltration, atrophy, and muscular necrosis from day 5 onwards.

## 3. Discussion

There is no specific antiviral drug treatment for chikungunya, and it is still a challenge on the clinical front. siRNA are ideal chemically synthesized drug candidates and can directly work on a target gene in a sequence-dependent manner. Since siRNAs are polyvalent anionic and highly hydrophilic mid-sized molecules, delivery of these molecules into cells is very difficult. They are easily degraded by nucleases in the blood, resulting in poor accumulation of siRNA in a target tissue. Hence, it is crucial to find a proper drug-delivery system for the development of siRNA-based drugs. Lipid-based nanoparticles are a suitable carrier for drug and nucleic-acid delivery. They have low toxicity and immunogenicity and excellent biocompatibility, biodegradability, structural flexibility, and ease of large-scale preparation. Several lipid-based formulations have been permitted and are being used around the world for several disease treatments. In this study, we prepared four different formulations using various cationic lipids to evaluate and rationalize an optimum delivery system that can show maximum siRNA complexation with a sufficient cationic surface charge. Previously, we reported that ODA itself has the potential to inhibit the CHIKV [26]. In the case of four formulations, earlier reports have indicated nanoparticles of PLGA-polyethylene glycol (PLGAPEG), with a strong potential to cross the blood–brain barrier and used to cater drug payloads [27,28]. It is highly uncertain that siRNA can cross the cell membrane due to its negative charge. However, a few reports indicate that nanoparticles facilitate siRNA internalization via endocytosis and as reasonable vectors for siRNA payloads owing great protection to the siRNA before the delivery of siRNA to the target [29]. Cationic lipids used can lockdown the negatively charged siRNA molecules inside the nanoparticles through electrostatic interactions.

Four delivery systems were evaluated for their toxicity, and F3 and F4, having a CC-50 value >100 µg/mL, were selected for further comparison of their in vitro CHIKV inhibition potential. Among F3 and F4 delivery systems, F3 indicated weak cationic charges and did not demonstrate high complexation with siRNA; therefore, it did not effectively downregulate the target genes in vitro. Further agarose-gel-electrophoresis results showed retardation of siRNA movement by siRNA-F4 SLN compared to naked SiRNA, which indicated the strength of complexes by exhibiting stable complexation. Hence, the F4 delivery system was selected for siRNA and further evaluation of their in vitro and in vivo anti-CHIKV potential.

Formulation F4 prepared using ODA acted as a cationic charge inducer in the protonated form and strongly interacted with the negatively charged siRNA, thereby facilitating the self-assembly of the delivery-system components. Earlier reports of octadecylamine-based liposomes indicated strong inhibition (80%) against baculovirus (BV) in a dose-dependent manner. The binding of ODA liposomes to the cell membranes was high and was not cytotoxic to normal cells. Aditionally, ODA liposomes indicated good antiviral effects on herpes simplex virus type 1 in A549 cells in comparison to acyclovir [30]. In another study, Chitosan oligosaccharide-SS-Octadecylamine (CSSO), a redox-responsive nano-sized polymeric carrier was developed that can entrap DrzBC and DrzBS (10–23 DNAzyme), and it could block the expression of HBV e- and s- genes, respectively. In comparison to Lipo2000, the polymeric carrier can be as efficient as anti-hepatitis B virus gene therapy [31]. Therefore, the combinatorial effects of ODA with siRNA cationic nanoparticles showed enhanced growth inhibition of CHIKV suggesting their potential advantages in clinical settings.

From ODA, we observed a maximum of 45.7 ± 4.2 mV for the surface charge, and a further increase in ODA concentration did not increase the charge proportionally. With the direct-method complexation of cationic lipids with siRNA, the particle size was observed at a maximum at an N/P ratio of 6, and the surface charge increased from 20 to 45 mV as the N/P ratio increased from 1 to 8. The results were in good correlation with previously reported data [32,33,34]. The in vitro results were in a similar trend to those earlier reported for anti-miR-191 delivery using ODA-based liposomes [35]. Earlier reports indicated that ODA-based nanoparticles aided better stability, minimizing the leakage of entrapped drugs with a controlled release [36]. Studies even found that ODA-based lipid systems have greatly enhanced the delivery and bioavailability of several anticancer drugs [37,38,39]. Thus, we can affirm that ODA-based nanolipid formulation exhibits a combinatorial nature by CHIKV inhibition at a higher concentration aiding for effective transfection of siRNA at a lower concentration.

In vitro study results indicated more significant inhibition of viral load by siRNA loaded with the F4 SLN delivery system in Vero cells compared to the inhibition shown by naked siRNA transfected by the use of lipofectamine. In vivo anti-CHIKV activity of the siRNA-loaded F4 SLN delivery system was further evaluated in the in vivo system using C57BL/6 mice. We evaluated serum and skeleton muscle of infected feet for detection of the presence of CHIKV, which corresponds to the CHIKV infection in humans. Treatment with siRNA complexed with F4 SLN exhibited a significant reduction in the serum viral load as evaluated at days 5 and 7 by real-time PCR. The combination of Chik-1 and Chik 5 delivered with F4 showed a ~99% reduction in viremia compared to the virus control group at days 5 and 7 in both serum and muscle tissue. The results were further validated by histopathology, which found that treatment of the siRNA F4 complex reduced the inflammatory cell infiltration, atrophy, and muscular necrosis from day 5 onwards.

Our results indicate that siRNA complexed with the F4 delivery system inhibited the viral replication in infected Vero cells as well as decreased the viral burden and helped ameliorate acute disease symptoms in CHIKV-infected mice.

The study concludes that cationic lipids, such as octadecylamine with ease of synthesis and characterization, indicated maximum complexation by structural condensation of siRNA owing to high transfection alone and synergistic inhibition of CHIKV along with siRNA in both in vitroand in vivo models. Therefore, octadecylamine-based cationic lipid nanoparticles can be embraced and explored as safe, potent, and efficient nonviral vectors overcoming siRNA in vivo complexities against chikungunya. In the future, nanoparticles containing the siRNA approach can be used in developing a delivery system for the treatment of other viral disease treatments.

## 4. Materials and Methods

### 4.1. Materials

Octadecylamine (stearylamine),C9 lipid (composition from Dr. Surendar Reddy, IICT), dioctadecylamine, branched PEI-g-PEG -PEG Mn 5000; 18:1 (Δ9-Cis) PC–1,2-dioleoyl-sn-glycero-3-phosphocholine (DOPC), 1,2-dihexadecanoyl-sn-glycero-3-phosphocholine (DPPC), Resomer^®^ R 202 S, poly (D, L-lactide) (PLGA 50:50), dimethyldioctadecylammonium bromide (DDAB), polyvinyl alcohol (85–90% hydrolysed) (PVA), 1,2-distearoyl-sn-glycero-3-phosphoethanolamine-N-[amino(polyethylene glycol)-2000] (ammonium salt), 1,2-distearoyl-sn-glycero-3-phosphoethanolamine-N-[methoxy(polyethylene glycol)-2000] (ammonium salt), powder (18:00 PEG 2000 PE), chloroform (DSEP PEG 2000), glycerylmonostearate (GMS), Tween 80, cholesterol, lecithin, dicholoromethane (DCM), and chloroform from Sigma Aldrich (St. Louis, MO, USA) were purchased.

### 4.2. Vero Cells and Virus Strains

A vero cell line was maintained using MEM (Himedia, Mumbai, India), supplemented with 10% FBS (Gibco, Grand Island, NY, USA) and antibiotic-antimycotic (Sigma-Aldrich, St. Louis, MO, USA) at 37 °C and 5% CO_2_. Chikungunya (CHIKV, strain no. 061573, P-2, African genotype) were used for this study. CHIKV stock was propagated in Vero cells at ICMR-NIV, Pune, and stored at −80 °C.

### 4.3. Animals, Housing, and Diets

C57BL/6 mice (3–4 weeks), which were bred in-house, were used for the in vivo experiments. Animals were maintained in the BSL-2 facility with a controlled temperature of 23 ± 2 °C and 40–70% relative humidity with a 12 h light–dark cycle. Animals had free access to a standard 9-pellet diet and fresh water. Animals were allowed to acclimatize for one week before the experimentation. All procedures described were reviewed and approved by the Institutional Animal Ethics Committee (IAEC), National Institute of Virology (NIV), Pune, India (IAEC number CHK501 approved on 20 December 2017). The animal experiments were conducted in accordance with the Committee for the Purpose of Control and Supervision of Experiments on Animals (CPCSEA) guidelines.

### 4.4. siRNA

Two effective siRNAs, i.e., CHIK1 and CHIK 5 (designed earlier against the ns1 and E2 gene regions), were used in this study [7]. These siRNAs showed significant inhibition against CHIKV in the in vitro system and even in mice showing complete inhibition of CHIKV [7].

### 4.5. Formulation of Different Delivery Systems

For effective siRNA delivery, four different nano delivery systems were prepared using different cationic lipids by distinct techniques. The prepared individual delivery systems were designated as F1, F2, F3, and F4, respectively.

#### 4.5.1. Preparation Method for F1 and F3 Delivery Systems

Both F1 and F3 delivery systems were hybrid polymeric lipid nanoparticles prepared by the double-emulsion solvent-evaporation method [40]. The F1 and F3 delivery systems had PEI-PEG (5 mg) and dioctadecylamine (3 mg) as cationic lipids, respectively, along with other components (Table 1). Briefly, all the required PEI-PEG, PLGA 50:50 (F1), dioctadecylamine, PLGA 50:50 (F3), DDAB, DOPC, DPPC, 18:00 PEG 2000 PE, cholesterol, and lecithin were accurately weighed and dissolved in 1 mL of DCM. The organic phase was slowly injected into 3 mL cold polyvinyl alcohol PVA (0.5% *w*/*v*) under probe sonication (Sonics Vibra cell, Newtown, CT, USA) at 50% amplitude for 15–20 min, forming a primary emulsion. The primary emulsion was added drop-wise to 10 mL cold PVA (0.5% *w*/*v*) under constant stirring at 900 rpm, leaving it for 30 min to form a double emulsion. Further, the emulsion was probe-sonicated at 50% amplitude for 25 min in an ice container. The resultant emulsion was left under stirring at 1200 rpm for 3 h at room temperature to evaporate the chloroform. The prepared formulations were stored in a sealed container at 4 °C.

#### 4.5.2. Preparation Method for F2 and F4 Delivery Systems

We determined that 8 mg of ODA was capable of inducing a firm cationic surface charge in comparison to 2, 4, 6, and 8 mg of ODA. Therefore, 8 mg of ODA was used as an optimal lipid concentration. Both the F2 and F4 delivery systems are based on SLN prepared by the emulsification-solvent evaporation method [41,42,43]. The F2 and F4 delivery systems had new C9 lipids (9 mg) and ODA (8 mg) as cationic lipids, respectively, along with glyceryl monostearate (GMS), cholesterol, DOPC, and DPPC (helper lipids). Briefly, for preparing both F2 and F4, a premixed solution of lipids in 1 mL of chloroform (organic phase) (Table 2) was added to 0.5% of Tween 80 (aqueous phase) and homogenized (Ultra Turrax, Reichenbach, Germany) for 5 min and sonicated at a 50% amplitude for 20 min and kept in stirring condition for 2–3 h until the solvent evaporated. The prepared formulations were stored in a sealed container at 4 °C.

### 4.6. Particle Size, Polydispersity, and Determination of Zeta Potential

The average particle size, surface charge, and size distribution of developed delivery systems were determined by dynamic light scattering (DLS) using a Zetasizer Nano ZS (Malvern Instruments, Malvern, UK). ODA-SLNs surface charge was elucidated by investigating the ζ-potential using the earlier reported method [44]. All measurements were performed using deionized water at 25 °C in triplicate (*n* = 3) in a 1:50 dilution.

### 4.7. Determination of Optimal Lipid-to-siRNA Complexation Ratio for F4

The particle size and the Zeta potential were used to determine the siRNA/optimal lipid nitrogen/phosphate (N/P) complexation ratio. The ODA-based lipoplex (F4) was prepared at different (N/P) atomic ratios that were tested: 1, 2, 4, 6, 8 and 10, keeping the siRNA quantity constant at 1000 pmol. The lipid nanoparticle size and charge were observed using a Zetasizer Nano ZS (Malvern Instruments, Malvern, UK) [45].

### 4.8. Cytotoxicity Assay (MTT Assay)

The in vitro cytotoxicity of four delivery systems (F1, F2, F3, and F4) on Vero cells was evaluated using a 3-(4,5-dimethythiazol-2-yl)-2,5-diphenyl tetrazolium bromide (MTT) reduction assay [46]. Briefly, monolayers of Vero cells in 96 well plates were incubated with different concentrations of test formulations from 0 to 200 µg/mL for 48 h at 37 °C, incubated with a MTT solution (5 mg/mL) for an additional 3 h at 37 °C. The solubilized formazan crystals were measured using a microplate reader (BioTek Synergy HT, Winooski, VT, USA) at 570 nm with a reference filter of 690 nm. The percentage cell viability was calculated in comparison with control cells treated with the same volume of sterile PBS using GraphPad Prism 5.0 software [47].

### 4.9. Formation of siRNA-SLN Complexes

The siRNA-SLN complexes were prepared by incubating1000 pmol of the siRNA with 50 µg/mL of nanoparticulate formulation at 4 °C for 4 h. The total-lipid concentration needed to complex with siRNA was maintained with no toxic effect on the Vero cells. The siRNAdelivery complex was further evaluated for stability.

### 4.10. Determination of siRNA F4 SLN Complexes’ Stability by Gel Retardation Assay

The complexation stability of the F4 delivery systems with the siRNA was analyzed using agarose-gel electrophoresis. The complexation of delivery systems with siRNA was carried out using different temperatures, lipid concentrations, and siRNA. Proper complexation was allowed by the incubating delivery system with siRNA for about 4 h at 4 °C. Gel electrophoresis was performed using 1% agarose gel at 65 V for 45 min in a TAE buffer solution [43].

### 4.11. In Vitro Release Studies

The siRNA release from the optimized ODA-SLN formulation (F4) was studied in phosphate buffer saline PBS (pH 7.4) and acidic conditions (pH 6.4). Formulations (4 mL) were centrifuged at 13,000× *g* for about 28 min at 37 °C in RNase-free eppendorfs. The obtained pellets were re-suspended in PBS and acidic solutions (3 mL). Further, the pellets were stirred at 100 rpm for 7 days at 37 °C. At different time points, the samples were aliquoted and were centrifuged at 13,000× *g* for 28 min at 25 °C [45]. The collected supernatant with an equal volume of fresh PBS or acidic solution was added to the sample. The siRNA concentration obtained in the supernatant was analyzed using a Thermo Scientific NanoDrop 1000 Spectrophotometer (Waltham, MA, USA) [48].

### 4.12. Determination of In Vitro Anti-CHIKV Activity of the siRNA-Loaded SLN Delivery System

Vero cells were infected with CHIKV and after one hour treated with naked siRNAs (Chik 1 and Chik 5) and siRNAs loaded with delivery systems individually and in combination at 37 °C. CHIKV replication inhibition was determined by quantitative RT-PCR (qRT-PCR) and focus forming units (FFU) assay at 24 and 48 h p.i. [49]. For the FFU assay, different dilutions of tissue-culture supernatants of infected and siRNA transfected cells were added to a monolayer of Vero cells and incubated at 37 °C for 1 h. After the incubation, the medium was replaced by an overlay medium and was further incubated for 24 h. The cells were fixed, and primary and secondary antibodies were added, followed by addition of true-blue peroxidase substrate. Cells were washed with PBST and were incubated at 37 °C between each step of additions. Foci were obtained and calculated as FFU per mL. For the qRT-PCR assay, RNA extraction from the cells was done using QIAmp viral RNA minikit (QIAGEN, Valencia, CA, USA) method. Total viral RNA was determined by real time-PCR using the standard-curve method. The primers and the probe targeted the E3 gene, and the sequences have been reported earlier [49]. In addition, the IFA assay for the quantitative estimation of virus infectivity as described earlier was also performed [49]. In brief, for IFA treated cells, cover slips were fixed, followed by addition of a primary antibody (immune mouse serum against a CHIKV clone) and a secondary (anti-mouse IgG FITC conjugate produced in goat) conjugate antibody. Cells were then mounted with a mounting solution containing DAPI (nuclear stain) and were observed under an EVOS Floid Cells imaging station microscope with 20× fixed magnification. The percent cells infection was calculated by Image J software.

### 4.13. Determination of In Vivo Anti-CHIKV Activity of the siRNA-Loaded SLN Delivery System

To evaluate the in vivo antiviral activity of the siRNA-loaded F4 SLN formulation, C57BL/6 female mice were infected with CHIKV using the intramuscular route (i.m.) (100 µL of 10^7^ pfu/mL), and mice were treated with different siRNA- (dose 1 mg/kg of body weight) loaded F4 formulations via the intravenous (i.v.) route on day 3 pid. Each mousereceived almost 50 µL of siRNA- (25 µg) loaded F4 SLN (3.75 mg/mL). Submandibular blood was collected at 3, 5, and 7 pid for viremia analysis. At each time point, mice were euthanized by cervical dislocation. Muscle tissue from hind feet from mice were collected; histopathology and RT-PCR analysis was performed at different time points [49].

## Figures and Tables

**Figure 1 molecules-27-01170-f001:**
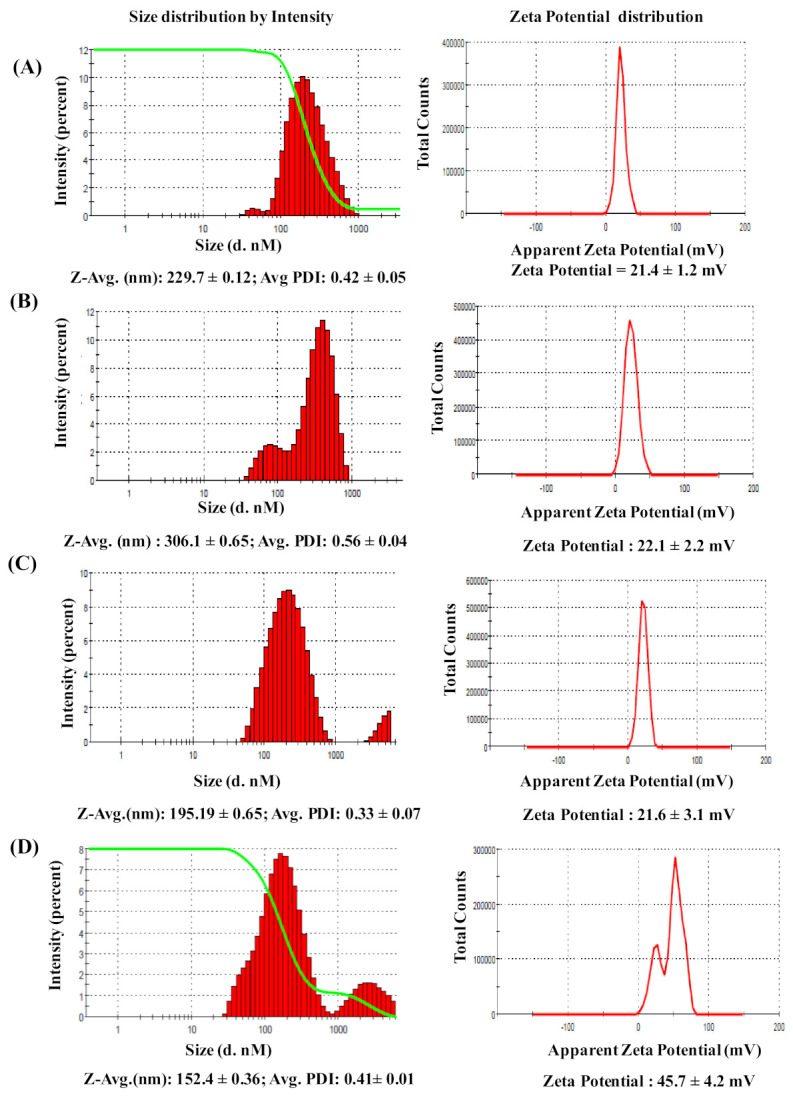
Dynamic light scattering (DLS) particle-size distributionsby intensity of experiments and Zeta potential of finally optimized (**A**) F1, (**B**) F2, (**C**) F3, and (**D**) F4 formulations measured by a Malvern Zetasizer Nano ZS. The average particle size in nm and polydispersity index (PDI) and Zeta potential in mV were shown for each formulation. All experiments were performed in triplicates; values are expressed as mean ± SEM. Besides, F1, F3 and F4 nanoparticles showed well polydispersity with PDI < 0.5. Zeta-potential analysis showed that F4 exhibited an indicated maximum cationic charge density (45.7 ± 4.2) in comparison to other formulations with average size (152.4 ± 0.36), which was better in comparison to all other formulations. It is also reported that nanoparticles with a Zeta potential above (+/−) 30 mV are stable in suspension.

**Figure 2 molecules-27-01170-f002:**
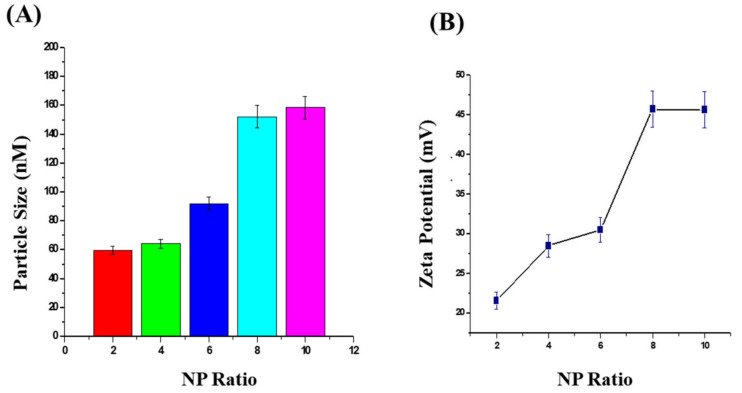
Representation of the trend of solid lipid nanoparticles particle size (nm) (**A**) and Zeta potential with N/P ratio at fixed siRNA concentration (**B**). All the values are expressed as mean ± SEM (*n* = 3).

**Figure 3 molecules-27-01170-f003:**
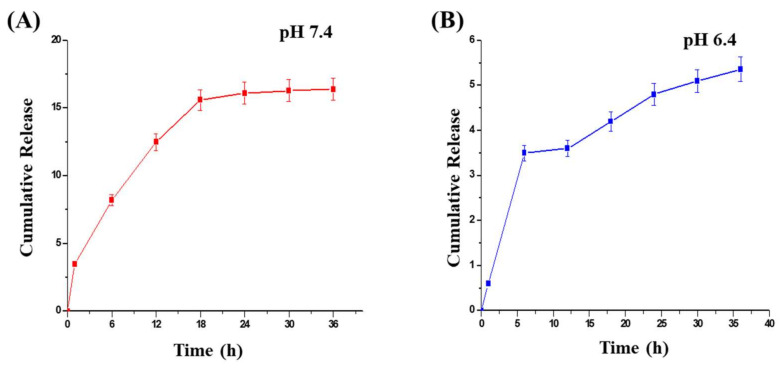
In vitro release profile of F4 formulation at different pH: (**A**) pH 7.4 and (**B**) pH 6.4; all the values are expressed as mean ± SEM (*n* = 3).

**Figure 4 molecules-27-01170-f004:**
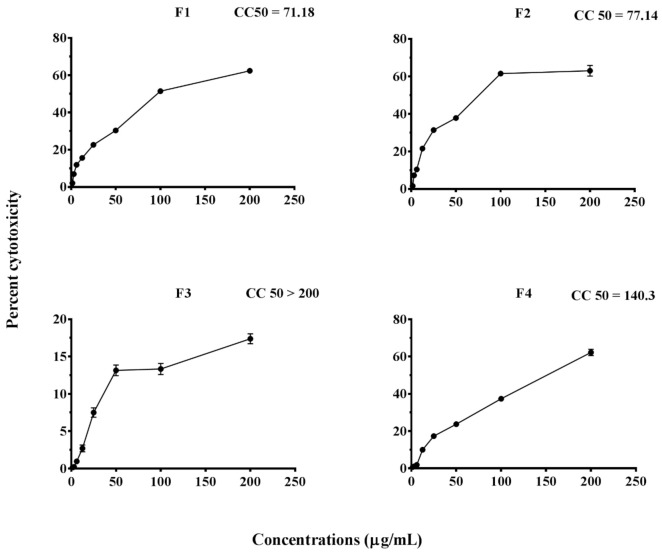
Cell cytotoxicity of different formulations against Vero E6 cells at various concentrations using an MTT assay. All the values are expressed as mean ± SEM (*n* = 3).

**Figure 5 molecules-27-01170-f005:**
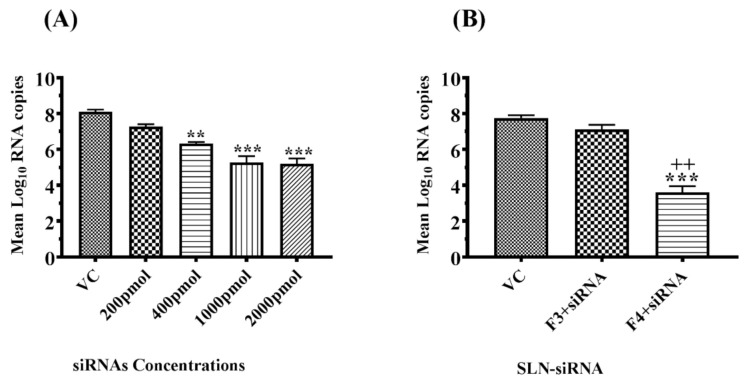
(**A**) Effect of transfection of different siRNA concentrations using lipofectamine reagent against CHIKV at 24 h treatment using qRT-PCR analysis, (**B**) Transfection efficiency comparison between F-3 and F4 RT-PCR results. All the values are expressed as mean ± SEM (*n* = 3); ** *p* < 0.01, *** *p* < 0.001 vs. VC, virus control group; ++ *p* < 0.01 vs. F3 + SiRNA treated group.

**Figure 6 molecules-27-01170-f006:**
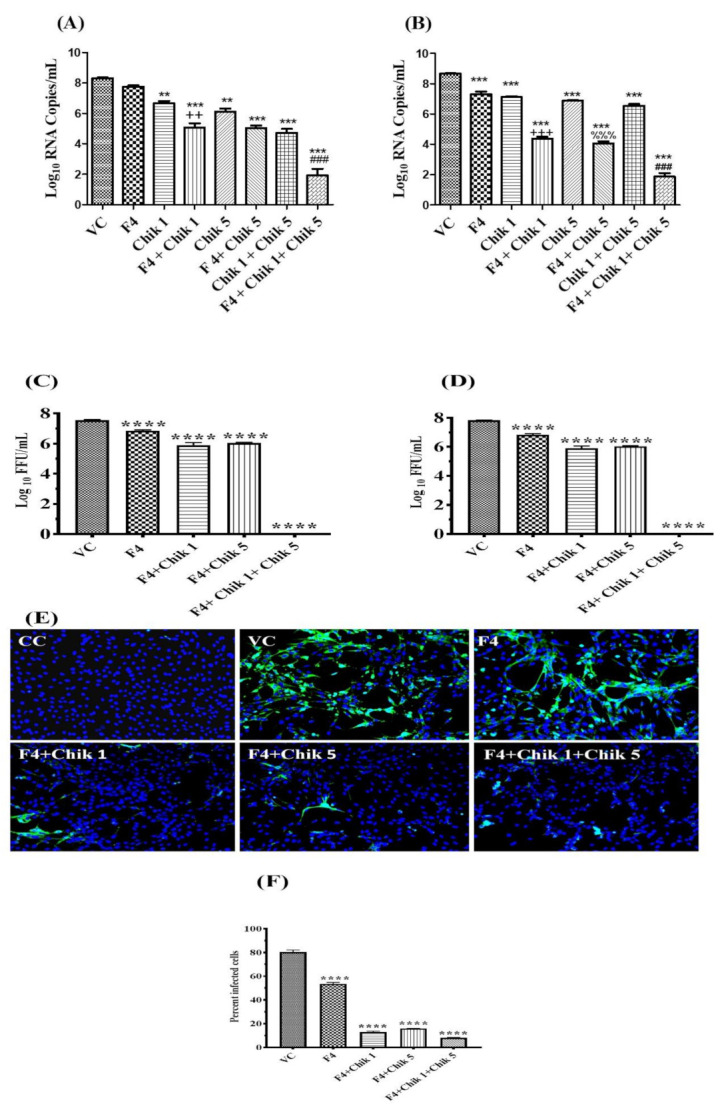
Effect of siRNA treatment against CHIKV replication. Naked siRNA (transfected with lipofectamine) and siRNA conjugated with F4 SLN effect at 24 h (**A**) and 48 h (**B**) using qRT-PCR analysis; siRNA conjugated with F4 SLN formulation using focus forming units (FFU) assay at 24 h (**C**) and 48 h (**D**); immunofluorescence assay (24 h treatment); (**E**) and percentage of infected cells (**F**).All values are given as mean log10 RNA copies/well ± SEM; **** *p* < 0.0001,*** *p* < 0.001, ** *p* < 0.01 vs. VC, virus control. ++ *p* < 0.01, +++ *p* < 0.001 vs. Chik 1, ^%%%^ *p* < 0.001 vs. Chik 5, ^###^ *p* < 0.001 vs. Chik 1 + Chik 5. All the siRNA combination conjugated with F4 SLN showed significant (*p* < 0.001) reduction to their respective treatment of naked siRNA with lipofectamine at 48 h; it indicated the superiority of the F4 SLN delivery system over transfection with lipofectamine.

**Figure 7 molecules-27-01170-f007:**
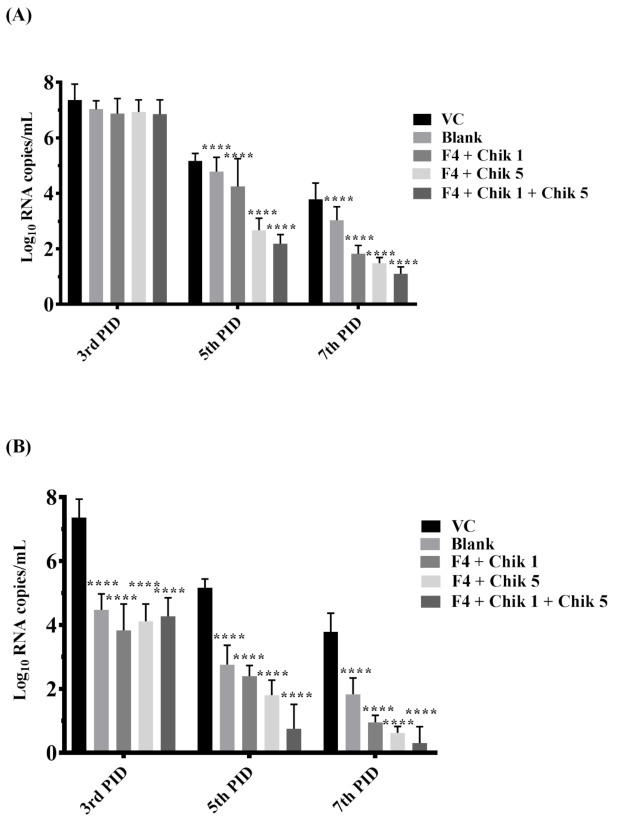
In vivo anti-CHIKV activity of siRNA-complexed SLN delivery system. The reduction in CHIKV copies/mL in (**A**) serum and (**B**) muscle tissue after treatment with the siRNA-loaded F4 delivery system. C57BL/6 mice (n *=* 9) were infected with CHIKV intramuscular route (100 µL of 10^7^ pfu/mL). After 72 h p.i. (3rd day), mice were treated with the siRNA-loaded F4 delivery system by intravenous route with different treatment combinations. Viral RNA copies were checked in serum and muscle tissue at different time intervals. CHIKV E3 RNA was quantitated using real-time RT-PCR. Values are given as LOG10 RNA copies/mL serum and LOG10 RNA copies/mg of tissue. **** *p* < 0.0001 VC, virus control.

**Figure 8 molecules-27-01170-f008:**
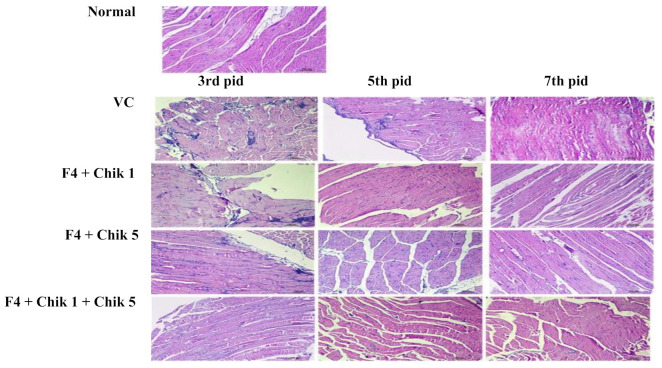
Histopathological changes in mouse muscle tissues after chikungunya infection and siRNA-loaded F4-delivery-system treatment.

**Table 1 molecules-27-01170-t001:** Compositions F1 and F3 delivery systems.

F1	F3
PLGA 50:50–3 mg PEI-PEG 5000–5 mg DDAB–5 mg Lecithin–2 mg PVA–0.3% (3 + 10 mL)	PLGA 50:50–3 mg Dioctadecylamine–3 mg DOPC–5 mg DPPC–5 mg Cholesterol–2 mg 18:00 PEG 2000 PE–2 mg PVA–0.3% (3 + 10 mL)
Total volume: 14 mL Total lipids: 15 mg	Total volume: 14 mL Total lipids: 20 mg

**Table 2 molecules-27-01170-t002:** Compositions of the F2 and F4 delivery systems.

F2	F4
C9 lipid–9 mg DOPC–5 mg DDAB–5 mg Cholesterol–5 mg DSPE-PEG 2000–3 mg Tween 80 (0.5%)–10 mL	Stearylamine–8 mg GMS–60 mg DOPC–5 mg DPPC–5 mg DSEP PEG 2000–2 mg Cholesterol–2 mg Tween 80 (0.5%)–10 mL
Total volume: 11 mL Total lipids: 27 mg	Total volume: 11 mL Total lipids: 82 mg

## Data Availability

Not applicable.

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
