# Peer review of "Effect of Cationic Lipid Nanoparticle Loaded siRNA with Stearylamine against Chikungunya Virus"

_molecules, 2022, doi:10.3390/molecules27041170_

Round 1

Reviewer 1 Report

In this manuscript, the authors describe the use of siRNA-loaded solid lipid nanoparticles (SLNs) for inhibition of Chikungunya (CHIKV) virus. Overall, they demonstrate that their “F4” lipid formulation containing the cationic lipid stearylamine (SA) can effectively encapsulate CHIK1 and CHIK5 siRNA and inhibit CHIKV replication in vitro and in vivo. Publication in Molecules is recommended pending the following revisions:

  • The authors mention that SA itself inhibits CHIKV; however, SA-mediated inhibition has not been directly compared to the F4 SLN formulation. This would be an important control to include to demonstrate that siRNA-loaded F4 SLNs are beneficial over SA only. This comparison will also support the need for siRNA to treat CHIKV, rather than
  • Last paragraph of introduction: Please cite the manuscript that defines CHIK1 and CHIK5 siRNA sequences.
  • Results 2.2 (Particle size, polydispersity, and zeta potential determination): Size data for the F4 lipid formulation depicts a bimodal population. Can the authors comment on why they see this trend? If filtration is used to remove large aggregates, is the efficacy of the siRNA-loaded F4 SLN affected?
  • Results 2.3 (Evaluation of optimal lipid to siRNA complexation ratio for F4): siRNA release is low under the conditions used in this assay, as expected. Can the authors comment on the importance of these results and justify why pH 7.4 and 6.4 were utilized in this assay?
  • Results 2.5 (Stability studies of developed lipid delivery systems): Data to support stability studies not shown, and methodology related to stability studies not explained. Please include experimental details in the methods section and consider adding a supplemental figure to depict sIRNA stability.
  • Figure 6: define “FFU assay” in figure legend. It would also be helpful to update figure labels so that it is clear which plots are for 24 h vs. 48 h treatment. In addition, it would be beneficial to compare statistical significance of siRNA knockdown between lipofectamine-siRNA to SLN-siRNA, rather than comparing to VC control, in order to show that the SLN-siRNA formation is more efficacious than transfection.
  • Figure 7: Can the authors compare all data directly in one plot? It is difficult to visually observe conclusions and differences between treatment groups when each data set is represented separately and when the y-axis ranges are different.
  • Methods Section 4.8: Should “percent inhibition” be corrected to “percent cell viability” with regard to the MTT assay? Please describe how percent cell viability was calculated (i.e. compared to what control?)
  • Methods Section 4.10: Please define “coherence ability.” It is highly recommended that the authors include their gel electrophoresis image and analysis as a supplemental image to illustrate siRNA complexation with each delivery system. In addition, quantification methods should be described here.
  • Methods Section 4.11: Please explain why formulations were centrifuged at 13,000 g for 28 mins at 37°C prior to the in vitro release study.
  • Methods Section 4.12: Please elaborate on the real-time PCR protocol utilized in this manuscript. Specifically, include primers used to detect siRNA targets. As depicted, it is unclear if reduction in RNA quantity is a general reduction in viral RNA or if you are specifically detecting knockdown of ns1 and E2 mediated by CHIK1 and CHIK5 siRNA. Include detection probe sequences utilized, if relevant. It would also be helpful to provide a citation for the FFU assay and briefly explain the protocol in more detail (e.g. what is overlay medium?)
  • Methods Section 4.13: Please describe quantity of siRNA-SLN formulation injected per mouse.

Formatting suggestions:

  • Degree symbol for temperature is inconsistent in format throughout manuscript.
  • For figures with multiple plots depicting similar data, it would be helpful to make all y-axis ranges the same to more easily observe trends in data.
  • If error bars or p-values used, please define type of error calculated, number of trials conducted, and/or p-value ranges for all figures.

Author Response

Response to Reviewer’s Comments- (Manuscript ID: molecules-1535050)

Reviewer 1:

In this manuscript, the authors describe the use of siRNA-loaded solid lipid nanoparticles (SLNs) for inhibition of Chikungunya (CHIKV) virus. Overall, they demonstrate that their “F4” lipid formulation containing the cationic lipid stearylamine (SA) can effectively encapsulate CHIK1 and CHIK5 siRNA and inhibit CHIKV replication in vitro and in vivo. Publication in Molecules is recommended pending the following revisions:

The authors mention that SA itself inhibits CHIKV; however, SA-mediated inhibition has not been directly compared to the F4 SLN formulation. This would be an important control to include to demonstrate that siRNA-loaded F4 SLNs are beneficial over SA only. This comparison will also support the need for siRNA to treat CHIKV, rather than

Last paragraph of introduction: Please cite the manuscript that defines CHIK1 and CHIK5 siRNA sequences.

Response: We have used blank F4 SLN i.e. without siRNA loading for a comparison as control group, as F4 siRNA itself have SA as a component. The concentration of F4 SLN used in vitro was 10 µg/ml which contains the SA concentration approximately 0.97 µg (3.6 µM) (please refer table 2). It is clear by our results that siRNA loaded F4 SLN significantly reduced viral load in vitro at very less concentration of siRNA i.e. 400 pmole. In our previous research work the reported effective antiviral concentration of SA was 50 µM in vitro that is comparative higher and above this concentration SA was toxic for Vero cells. Hence the importance of siRNA delivery to treat chikungunya is self-explanatory.

Results 2.2 (Particle size, polydispersity, and zeta potential determination): Size data for the F4 lipid formulation depicts a bimodal population. Can the authors comment on why they see this trend? If filtration is used to remove large aggregates, is the efficacy of the siRNA-loaded F4 SLN affected?

Response: We thank reviewer for this concern. The figure depicted the size distribution of blank F4 formulation before complexation with siRNA. The larger size population may be due few macro particles or sometimes due to presence of any contaminants within the zeta sizer also. To remove such population, SLN formulation were filtered with 0.45 µM filter before conjugation with siRNA to ensure that such bimodal population would not be present in filtrate F4 SLN. (Page no. 3, Line no. 123-125)

Results 2.3 (Evaluation of optimal lipid to siRNA complexation ratio for F4): siRNA release is low under the conditions used in this assay, as expected. Can the authors comment on the importance of these results and justify why pH 7.4 and 6.4 were utilized in this assay?

Response: The objective of studying the release at pH 6.4 and pH 7.4 is to assess the pH dependent release of siRNA from the formulation. The results indicated that at the neutral condition the percent release of siRNA was higher and prolonged in comparison to the acidic condition. It should be noted that burst release observed at pH 6.4 was not seen in the release results obtained at pH 7.4 The in vitro results were in a similar trend to those earlier reported for anti-miR-191 delivery using SA-based liposomes. Earlier reports indicated that SA-based nanoparticles aided better stability minimizing the leakage of entrapped drugs with controlled-release.

Results 2.5 (Stability studies of developed lipid delivery systems): Data to support stability studies not shown, and methodology related to stability studies not explained. Please include experimental details in the methods section and consider adding a supplemental figure to depict siRNA stability.

Response: Details of stability studies and supplemental data file has been added to revised manuscript. (Page No.8, Line no. 229-232)

Figure 6: Define “FFU assay” in figure legend. It would also be helpful to update figure labels so that it is clear which plots are for 24 h vs. 48 h treatment. In addition, it would be beneficial to compare statistical significance of siRNA knockdown between lipofectamine-siRNA to SLN-siRNA, rather than comparing to VC control, in order to show that the SLN-siRNA formation is more efficacious than transfection.

Response: As reviewer suggested we have incorporated FFU full form in figure legend. We have also mentioned clearly about 24 and 48 hr. We agree with reviewer suggestion about siRNA knockdown between lipofectamine- siRNA to SLN-siRNA and accordingly we have modified Fig 6 and statement included in the manuscript (Page no. 8 & 9, Figure 6.).

Figure 7: Can the authors compare all data directly in one plot? It is difficult to visually observe conclusions and differences between treatment groups when each data set is represented separately and when the y-axis ranges are different.

Response: As suggested, we have modified the Fig 7.

Methods Section 4.8: Should “percent inhibition” be corrected to “percent cell viability” with regard to the MTT assay? Please describe how percent cell viability was calculated (i.e. compared to what control?)

Response: Thanks for suggestion, we have corrected statement and required details have been added in the same section (Page no.16, Line no. 540-548).

Methods Section 4.10: Please define “coherence ability.” It is highly recommended that the authors include their gel electrophoresis image and analysis as a supplemental image to illustrate siRNA complexation with each delivery system. In addition, quantification methods should be described here.

Response: Method section has been updated and supplemental gel electrophoresis image has been added. (Page no.17, Line no. 548-550).

Methods Section 4.11: Please explain why formulations were centrifuged at 13,000 g for 28 mins at 37°C prior to the in vitro release study.

Response: For conducting the in vitro release studies, the formulations were centrifuged in order to obtain the pellet containing siRNA complexed with lipid formulation. The pellet was re-suspended in buffer having pH 6.4 and 7.4. The speed and time of centrifugation were optimized at 13,000 g and 28 minutes respectively in order to avoid the loss of the formulation and to get maximum amount of pellet.

Methods Section 4.12: Please elaborate on the real-time PCR protocol utilized in this manuscript. Specifically, include primers used to detect siRNA targets. As depicted, it is unclear if reduction in RNA quantity is a general reduction in viral RNA or if you are specifically detecting knockdown of ns1 and E2 mediated by CHIK1 and CHIK5 siRNA. Include detection probe sequences utilized, if relevant. It would also be helpful to provide a citation for the FFU assay and briefly explain the protocol in more detail (e.g. what is overlay medium?)

Response: As suggested details of real time PCR have been incorporated in manuscript. Explanation have been incorporated. (Citation for FFU assay and brief explanation have been incorporated. (Revised section 4.12; Page no. 17; Line no. 574-593)

Methods Section 4.13: Please describe quantity of siRNA-SLN formulation injected per mouse.

Response: We have incorporated quantity of siRNA-SLN formulation injected per mouse in method section 4.13. (Page no.17, Line no. 581-594).

Formatting suggestions:

Degree symbol for temperature is inconsistent in format throughout manuscript.

Response: Corrected

For figures with multiple plots depicting similar data, it would be helpful to make all y-axis ranges the same to more easily observe trends in data.

Response: As suggested we have modified figures.

If error bars or p-values used, please define type of error calculated, number of trials conducted, and/or p-value ranges for all figures.

Response: As suggested we have added figure legends and above mentioned information also incorporated.

Reviewer 2 Report

This manuscript developed a cationic charged system for the effective complexation and delivery of siRNA against CHIKV using various lipids. Four formulated delivery systems were characterized and evaluated in an in vitro system. This manuscript shows that cationic lipid nanoparticles based on stearamine can be used as a safe, effective, and effective non-viral vector to overcome the complexity of siRNA targeting Chikungunya fever in vivo. However, there are some serious issues in interpretation. The detailed comments are as follows.
  1. The authors need to elaborate on the novelty of the reported work in the introduction part.
  2. Octadecylamine (Stearylamine) is more commonly abbreviated as ODA than SA.
  3. The method of calculating IC50 is not described in detail, and the value of IC50 is not marked.
  4. The qualities of figures should be improved, Figure 1 seems like a table, figure 4, Figure 7, Figure 9 should be rearranged, and the graphics should not obscure the relevant description. In addition, how siRNA is loaded on solid lipid nanoparticles should be more clearly reflected in the figure 9. Pay attention to the figure 2 caption.
  5. The SEM/TEM pictures are required as the authors talked about prepared NPs,
  6. The abscissa of the quantitative diagram in Figure 6D should not be the concentration.
  7. In manuscript 2.6, the author said, “the developed delivery systems F3 and F4 having IC-50 value >100 µg/ml, were considered non-toxic to Vero cells”, but the cytotoxicity of F3 in Figure 4 was inconsistent with that described by the author.
  8. The structure-property relationship need to be discussed in detail
  9. The reference citations should be improved with the latest studies
  • In the manuscript, the author refers to reference No.18 when explaining the significance of cationic lipids obtained in recent years. It is a 1995 article, which can not well summarize the progress of cationic lipids in recent years.
  • Some Format problem
(1) The cell viability in Figure 4 lacks units.
(2) In Figure 5, the representation method of analyzing significant differences should be consistent.
(3) The unit in the text should be “mL/µL” instead of “ml/µl.”
(4) There were 2.3 and 2.5 in the results, and 2.4 was missing.
(5) The picture layout and resolution in this article need to be handled carefully.
(6) The citation format of reference No. 26-17, 31-33 is wrong.
(6) The format of references is not standardized, some references are outdated, and there are few references in recent 3 years.
Please find out and correct it carefully.

Author Response

Response to Reviewer’s Comments- (Manuscript ID: molecules-1535050)

Reviewer 2:

This manuscript developed a cationic charged system for the effective complexation and delivery of siRNA against CHIKV using various lipids. Four formulated delivery systems were characterized and evaluated in an in vitro system. This manuscript shows that cationic lipid nanoparticles based on stearylamine can be used as a safe, effective, and effective non-viral vector to overcome the complexity of siRNA targeting Chikungunya fever in vivo. However, there are some serious issues in interpretation. The detailed comments are as follows.

  1. The authors need to elaborate on the novelty of the reported work in the introduction part.

Response: Novelty mentioned in the Introduction (Page no. 3, Line no. 99-100).

  1. Octadecylamine (Stearylamine) is more commonly abbreviated as ODA than SA.

Response: As suggested we have mentioned ODA for Octadecylamine (Stearylamine) rather than SA in revised manuscript.

  1. The method of calculating IC50 is not described in detail, and the value of IC50 is not marked.

Response: The mentioned IC50 should be CC50 which we have modified in revised version and method of calculation included and value of CC50 is marked in figure 4.

  1. The qualities of figures should be improved, Figure 1 seems like a table, figure 4, Figure 7, Figure 9 should be rearranged, and the graphics should not obscure the relevant description. In addition, how siRNA is loaded on solid lipid nanoparticles should be more clearly reflected in the figure 9. Pay attention to the figure 2 caption.

Response: We have modified figures as suggested and caption for figure 2 changed. As siRNA loaded on solid lipid nanoparticles reflection is not clear in figure 9, so removed from manuscript.

  1. The SEM/TEM pictures are required as the authors talked about prepared NPs,

Response: We do agree SEM and TEM experiments needs to be performed. As conducting these studies is beyond the scope of the present work, in the ongoing COVID crisis situation.  In future we will consider this suggestion and for the further improvement of delivery system we will perform SEM/TEM Imaging.

  1. The abscissa of the quantitative diagram in Figure 6D should not be the concentration.

Response: Thank you for the observation and we have corrected the typographical error in figure 6D.

  1. In manuscript 2.6, the author said, “the developed delivery systems F3 and F4 having IC-50 value >100 µg/ml, were considered non-toxic to Vero cells”, but the cytotoxicity of F3 in Figure 4 was inconsistent with that described by the author.

Response: Thank you for indicating mistake in figure 4 and accordingly we have corrected the figure.

  1. The structure-property relationship needs to be discussed in detail.

Response: The delivery systems proposed in the present study are hybrid polymeric and lipid nanoparticle systems. These systems were prepared by double emulsion and evaporation method. These systems were complexed with siRNA. The incorporated stearylamine induces cationic charge and produces strong inhibition against CHIKV. In addition, cholesterol, a component of these formulations brings stearic hindrance and imparts stability to the formulations. These optimized formulations imparted necessary properties such as particle size, polydispersity index and surface charge which are essential for optimal complexation of lipid with siRNA.

  1. The reference citations should be improved with the latest studies

Response: Reference citation modified and removed old studies.

In the manuscript, the author refers to reference No.18 when explaining the significance of cationic lipids obtained in recent years. It is a 1995 article, which can not well summarize the progress of cationic lipids in recent years.

Response: Latest reference have been added.

Some Format problem

(1) The cell viability in Figure 4 lacks units.

Response: We have modified Figure 4.

(2) In Figure 5, the representation method of analyzing significant differences should be consistent.

Response: As suggested, we have modified figure 5.

(3) The unit in the text should be “mL/µL” instead of “ml/µl.”

Response: As suggested we have modified unit.

(4) There were 2.3 and 2.5 in the results, and 2.4 was missing.

Response: Thank you for indicating about 2.4 section which was somehow missed during formatting.

(5) The picture layout and resolution in this article need to be handled carefully.

Response: As suggested we have modified figures.

(6) The citation format of reference No. 26-17, 31-33 is wrong.

Response: As suggested references have been checked.

(6) The format of references is not standardized, some references are outdated, and there are few references in recent 3 years.

Please find out and correct it carefully.

Response: As suggested latest reference have been added (Reference nos.17, 18, 22-23, 26-27,33-34,36-42, 44,46- 49).

Round 2

Reviewer 1 Report

Thank you to the authors for answering the reviewer comments. I recommend this article for publication pending text edits to fix any grammatical errors or typos.

Author Response

Comments and Suggestions for Authors

Thank you to the authors for answering the reviewer comments. I recommend this article for publication pending text edits to fix any grammatical errors or typos.

Response: As suggested, we have tried to fix grammatical mistakes.

Reviewer 2 Report

The authors should carefully revise the paper.

The quality of the figures still required improvement. Authors should look at this paper for reference: Biomaterials 211 (2019) 68–80

The font sizes are different even in the same picture, text is unable to read…many more

The language is also needed to review carefully for example

In this study, nanodelivery systems (Hybrid polymeric/ Solid lipid nano- 16
particles), it should be hybrid, not Hybrid.

Author Response

Reviewer 2

Comments and Suggestions for Authors

The authors should carefully revise the paper.

Response: As suggested we have critically revised manuscript.

The quality of the figures still required improvement. Authors should look at this paper for reference: Biomaterials 211 (2019) 68–80

Response: As suggested we have tried to improve figures quality.

The font sizes are different even in the same picture, text is unable to read…many more

Response: Thank you for suggestion and accordingly we have corrected font sizes of figures.

The language is also needed to review carefully for example

In this study, nanodelivery systems (Hybrid polymeric/ Solid lipid nano- 16
particles), it should be hybrid, not Hybrid.

Response: As suggested we have reviewed and modified language accordingly.